# Methoxyhispolon Methyl Ether, a Hispolon Analog, Thwarts the SRC/STAT3/BCL-2 Axis to Provoke Human Triple-Negative Breast Cancer Cell Apoptosis In Vitro

**DOI:** 10.3390/biomedicines11102742

**Published:** 2023-10-10

**Authors:** Chih-Pin Liao, Ya-Chu Hsieh, Chien-Hsing Lu, Wen-Chi Dai, Wei-Ting Yang, Kur-Ta Cheng, Modukuri V. Ramani, Gottumukkala V. Subbaraju, Chia-Che Chang

**Affiliations:** 1Division of General Surgery, Department of Surgery, Kuang Tien General Hospital, Taichung 433401, Taiwan; liaochihpin@gmail.com; 2Doctoral Program in Translational Medicine, National Chung Hsing University, Taichung 402202, Taiwan; chlu@vghtc.gov.tw; 3Doctoral Program in Tissue Engineering and Regenerative Medicine, National Chung Hsing University, Taichung 402202, Taiwan; asdf789515@gmail.com; 4Department of Obstetrics and Gynecology, Taichung Veterans General Hospital, Taichung 407219, Taiwan; 5Doctoral Program in Biotechnology Industrial Innovation and Management, National Chung Hsing University, Taichung 402202, Taiwan; qqqchyi@gmail.com; 6Department of Life Sciences, National Chung Hsing University, Taichung 402202, Taiwan; 7Department of Biochemistry and Molecular Cell Biology, Taipei Medical University, Taipei 110301, Taiwan; ktbot@tmu.edu.tw; 8Department of Organic Chemistry, Andhra University, Visakhapatnam 530003, India; ramani_v@yahoo.com (M.V.R.); subbarajugv@gmail.com (G.V.S.); 9Graduate Institute of Biomedical Sciences, Rong Hsing Translational Medicine Research Center, The iEGG and Animal Biotechnology Research Center, National Chung Hsing University, Taichung 402202, Taiwan; 10Department of Medical Laboratory Science and Biotechnology, Asia University, Taichung 413305, Taiwan; 11Department of Medical Research, China Medical University Hospital, Taichung 404327, Taiwan; 12Traditional Herbal Medicine Research Center, Taipei Medical University Hospital, Taipei 110301, Taiwan

**Keywords:** methoxyhispolon methyl ether, hispolon, SRC, STAT3, BCL-2, apoptosis, triple-negative breast cancer

## Abstract

Triple-negative breast cancer (TNBC) is the most aggressive subtype of breast cancer with few treatment options. A promising TNBC treatment approach is targeting the oncogenic signaling pathways pivotal to TNBC initiation and progression. Deregulated activation of signal transducer and activator of transcription 3 (STAT3) is fundamental to driving TNBC malignant transformation, highlighting STAT3 as a promising TNBC therapeutic target. Methoxyhispolon Methyl Ether (MHME) is an analog of Hispolon, an anti-cancer polyphenol found in the medicinal mushroom *Phellinus linteus*. Still, MHME’s anti-cancer effects and mechanisms remain unknown. Herein, we present the first report about MHME’s anti-TNBC effect and its action mechanism. We first revealed that MHME is proapoptotic and cytotoxic against human TNBC cell lines HS578T, MDA-MB-231, and MDA-MB-463 and displayed a more potent cytotoxicity than Hispolon’s. Mechanistically, MHME suppressed both constitutive and interleukin 6 (IL-6)-induced activation of STAT3 represented by the extent of tyrosine 705-phosphorylated STAT3 (p-STAT3). Notably, MHME-evoked apoptosis and clonogenicity impairment were abrogated in TNBC cells overexpressing a dominant-active mutant of *STAT3* (STAT3-C); supporting the blockade of STAT3 activation is an integral mechanism of MHME’s cytotoxic action on TNBC cells. Moreover, MHME downregulated BCL-2 in a STAT3-dependent manner, and TNBC cells overexpressing BCL-2 were refractory to MHME-induced apoptosis, indicating that BCL-2 downregulation is responsible for MHME’s proapoptotic effect on TNBC cells. Finally, MHME suppressed SRC activation, while *v-src* overexpression rescued p-STAT3 levels and downregulated apoptosis in MHME-treated TNBC cells. Collectively, we conclude that MHME provokes TNBC cell apoptosis through the blockade of the SRC/STAT3/BCL-2 pro-survival axis. Our findings suggest the potential of applying MHME as a TNBC chemotherapy agent.

## 1. Introduction

Breast cancer is the most frequently diagnosed cancer in women worldwide and the second leading cause of cancer-related death in women [1]. Based on gene expression profiles, breast cancer can be divided into six subtypes: luminal A, luminal B, human epidermal growth factor receptor 2 (HER2)-positive, normal-like, basal-like, and claudin-low. Regarding disease prognosis, luminal A is the best, while basal-like shows the worst. Notably, most basal-like breast cancer patients belong to the group of triple-negative breast cancer (TNBC), which is featured by a lack of estrogen receptor (ER), progesterone receptor (PR), and HER2. TNBC accounts for 10–20% of all invasive breast cancers and is associated with high rates of tumor metastasis and cancer recurrence [2,3,4,5,6,7]. Local treatment is similar to other invasive breast cancer subtypes and includes surgery-mastectomy with or without adjuvant radiation or breast-conserving surgery followed by adjuvant radiation. As few molecular targets are available for targeted therapy, chemotherapy remains the primary treatment option for TNBC. Hence, the demand for effective TNBC therapeutics is still urgent [8,9,10]. Notably, alternative approaches for TNBC therapy focus on targeting the signaling pathways essential for driving and sustaining TNBC malignancy, including PI3K/AKT/mTOR, WNT/β-catenin, Hedgehog, Notch, NF-κB, and STAT3 [11,12,13,14].

STAT3 is a transcription factor essential for promoting cell proliferation, survival, stemness, and inflammation [15,16]. Classical STAT3 signaling is triggered by the binding of cytokines (primarily interleukin 6 (IL-6)) or growth factors (e.g., epidermal growth factor) to their cognate receptors on the cell surface to recruit and activate cytosolic non-receptor tyrosine kinases JAKs or SRC, respectively. In turn, the activated JAKs or SRC directly phosphorylate the tyrosine 705 residue of STAT3 (p-STAT3), causing the dimerization of two p-STAT3 molecules through reciprocal binding between the Src Homology 2 (SH2) domain and phosphorylated tyrosine 705. Afterward, the p-STAT3 homodimer is translocated from the cytosol to the nucleus to upregulate the transcription of an arsenal of downstream target genes such as *BCL-2* [17,18]. Several endogenous STAT3 inhibitors like the suppressor of cytokine signaling (SOCS) family members, SH2 domain-containing phosphatase (SHP)-1 and SHP-2, and protein inhibitors of activated STAT3 (PIAS3) quench STAT3 signaling to ensure transient activation of STAT3 in physiological settings [17]. In contrast, a broad range of human cancers are manifested by sustained activation of STAT3, as evidenced by high p-STAT3 levels in tumor tissues, which predicts poor prognosis in general [17]. For TNBC, constitutive STAT3 activation is a common molecular feature. It is fundamental to driving every aspect of TNBC initiation and malignant transformation, such as promoting TNBC cell proliferation and survival, migration and invasion, angiogenesis, chemoresistance, cancer stemness, and immune suppression [18]. Notably, growing preclinical evidence has demonstrated the therapeutic benefit of STAT3 inhibition on TNBC cells, confirming STAT3 as a promising TNBC drug target [18].

Hispolon is a polyphenolic component of the medicinal mushroom *Phellinus linteus*. Preclinical studies have uncovered Hispolon’s diverse pharmacological activities, including anti-inflammation, anti-oxidation, anti-virus, anti-diabetic, and anti-cancer [19,20]. Owing to Hispolon’s promising anti-cancer activity on varied human cancers, numerous Hispolon analogs generated by chemical modifications of Hispolon’s aromatic ring have been investigated for their potential anti-cancer effects and possible molecular targets, such as histone deacetylase and NF-κB, using in silico approaches [21,22,23,24]. Along this line, our laboratory has reported the first evidence about the in vitro anti-bladder cancer effect of Hispolon methyl ether (HME) [25] and the in vitro anti-colorectal cancer activity of Dehydroxyhispolon methyl ether (DHME), along with the underlying mechanisms of action [26,27]. Still, the anti-cancer properties with the underlying mechanisms of action for most Hispolon analogs remain to be explored.

Herein, we present the first report of the in vitro anti-TNBC effect of another Hispolon analog, Methoxyhispolon Methyl Ether (MHME). We demonstrated the cytotoxic effect of MHME on a panel of human TNBC cell lines with a higher potency than Hispolon. Mechanistically, we verified that MHME induces TNBC cytotoxicity by provoking apoptosis in TNBC cells by thwarting the SRC/STAT3/BCL-2 pro-survival signaling axis. Considering the fundamental contribution of aberrant STAT3 signaling to TNBC, our findings suggest the potential of MHME to be regarded as a constituent of TNBC chemotherapy regimens.

## 2. Materials and Methods

### 2.1. Chemicals

MHME was chemically synthesized as described in Balaji et al. [22] and prepared as a 100 mM stock solution in dimethyl sulphoxide (DMSO) for storage at −20 °C until use. Recombinant human interleukin 6 (IL-6) was acquired from PeproTech (Rehovot, Israel) and stored as 10 mg/mL solution in 1× phosphate-buffered saline (PBS) (VWR International; Radnor, PA, USA). DMSO, glutathione, and polybrene were acquired from Sigma-Aldrich (St. Louis, MO, USA). All chemicals required for cell culture were obtained from Gibco Life Technologies (Carlsbad, CA, USA).

### 2.2. Cell Culture

Human TNBC cell lines HS578T (American Type Culture Collection (ATCC) HTB-126™), MDA-MB-231 (ATCC HTB-26™), and MDA-MB-436 (ATCC HTB-130™) were used as the TNBC cell model in this study. HS578T cells were cultured in Dulbecco’s Modified Eagle’s Medium, while both MDA-MB-231 and MDA-MB-436 cells were cultured in Leibovitz’s L-15 medium. All culture media were supplemented with 10% fetal bovine serum and 1% penicillin–streptomycin. In addition, the culture media for both HS578T and MDA-MB-436 cells were replenished with 10 µg/mL insulin, and 16 µg/mL glutathione was particularly added to MDA-MB-436 cell culture. Cells were grown at 37 °C in a humidified environment with 5% CO_2_ for culturing HS578T cells or without 5% CO_2_ for the growth of MDA-MB-231 and MDA-MB-436 cells.

### 2.3. Cell Viability Assay

TNBC cells (7 × 10^3^ cells/well) seeded in 96-well culture plates were subject to treatment with MHME (0, 6.25, 12.5, 25, 50 μM) for 24 h and 48 h, followed by assessment of cell viability using CellTiter 96^®^AQueous One Solution Cell Proliferation Assay (MTS) assay (Promega; Madison, WI, USA) as previously described [25,26,27,28] (Appendix A).

### 2.4. Clonogenicity Assay

TNBC cells (4 × 10^5^ cells) were treated with MHME (0, 25, 50 μM) for 24 h, followed by seeding 2 × 10^2^ of MHME-treated cells onto 6-well plates to grow into colonies in drug-free media for 10~14 days. The TNBC colonies were revealed by 1% crystal violet staining, and then the numbers of colonies were scored as stated previously [25,26,27,28].

### 2.5. Apoptosis Assay

MHME-induced TNBC cell apoptosis was quantitatively determined using Muse^®^ Annexin V & Dead Cell Assay Kit (Millipore, Burlington, MA, USA) for the extent of Annexin V exposed on the cell surface by following our established protocol [25,26,27,28]. Briefly, TNBC cells (3 × 10^5^ cells/well) after 24 h treatment with MHME (0, 25, 50 μM) were resuspended by trypsinization and washed twice with 1× PBS, followed by 20 min incubation at room temperature in Annexin V & Dead Cell reagent (100 μL) in the dark. Afterward, cells were applied to the Muse^®^Cell Analyzer (Millipore; Burlington, MA, USA) to score the extent of Annexin V-positive (apoptotic) cell populations.

### 2.6. Establishment of Stable Clones

The construction strategies of the plasmids pBabe-HA-STAT3-C, pBabe-BCL-2, and pBabe-HA-v-src, which were designed for the ectopic expression of N-terminal hemagglutinin epitope (HA)-tagged dominant-active *STAT3* mutant (STAT3-C) [29], *BCL-2*, and N-terminal HA-tagged dominant-active *SRC* (*v-src*) [30], have been previously described in detail [25,26,27,28]. The preparation and infection of pBabe-derived retroviral particles and subsequent positive selection for virus-infected cells were executed in accordance with our established protocols [25,26,27,28]. The ectopic expression of HA-STAT3-C, BCL-2, or HA-v-src proteins in stable clones was verified by immunoblotting.

### 2.7. Immunoblottinghme

Immunoblotting was conducted as previously described [25,26,27,28]. Primary antibodies against cleaved PARP (#9541), HA-tag (#3724), phospho-Src (Y416) (#6743), phospho-STAT3 (Y705) (#9145), and Src (#2108) were bought from Cell Signaling Technology (Boston, MA, USA). Primary antibodies against α-tubulin (GTX112141), BCL-2 (GTX100064), and STAT3 (GTX104616) were obtained from GeneTex (Irvine, CA, USA). All secondary antibodies were acquired from Jackson ImmunoResearch Laboratories (West Grove, PA, USA).

### 2.8. Statistical Analysis

All data derived from three separate experiments were shown as the mean ± standard deviation. Student’s *t*-test analyzed comparisons between two individual groups. The statistically significant differences were recognized based on a probability value (*p*) lower than 0.05.

## 3. Results

### 3.1. MHME Is More Potent Than Hispolon to Exert In Vitro Cytotoxicity in TNBC Cells

The anti-TNBC potential of MHME was first addressed by evaluating its cytotoxic effect on a panel of TNBC cell lines, including Hs578T, MDA-MB-231, and MDA-MB-436. It is noted that MHME markedly reduced the viability of all TNBC cell lines in a dose-dependent manner, with an IC_50_ of 5.54 ± 0.47 μM, 8.93 ± 0.85 μM, and 5.56 ± 0.49 μM for HS578T, MDA-MB-231, and MDA-MB-436 cells, respectively, after MHME treatment for 48 h (Figure 1A). Furthermore, MHME’s cytotoxic effect on TNBC cells was illustrated by MHME’s potential to suppress the colony-forming ability of all tested TNBC cell lines. In particular, under 50 μM MHME treatment, the clonogenicity levels of HS578T, MDA-MB-231, and MDA-MB-436 cells were dropped to 2.73 ± 3.48%, 1.87 ± 1.30%, and 0.85 ± 1.48% of their respective drug-free controls (*p* < 0.001) (Figure 1B). On the contrary, TNBC cells appeared to be refractory to Hispolon-induced cytotoxicity. As shown in Figure 1C, the IC_50_ of Hispolon for HS578T and MDA-MB-231 after 48 h treatment were all higher than 50 μM, while they were around 25 μM for MDA-MB-436 cells, which was nearly 5-fold higher than that of HMME. Accordingly, these findings indicated that MHME exerts evident cytotoxicity against various TNBC cell lines, and its cytotoxic effect on TNBC cells is more potent than that of its parental molecule, Hispolon.

### 3.2. MHME-Elicited TNBC Cytotoxicity Involves the Induction of Apoptosis

We next explored the mode of MHME-induced TNBC cytotoxicity. Immunoblotting revealed a dose-dependent elevation of poly (ADP-ribose) polymerase (PARP) cleavage (c-PARP), a canonical marker of apoptosis [31], in all examined TNBC cell lines following MHME treatment, suggesting the induction of apoptosis (Figure 2A). Furthermore, MHME-induced apoptosis in TNBC cells was substantiated by the increase in the levels of Annexin V-positive (hence apoptotic) populations in all MHME-treated TNBC cell lines according to flow cytometry analyses. As shown in Figure 2B, the levels of apoptotic population in HS578T cells were markedly enhanced from 17.77 ± 0.21% in drug-free control to 96.63 ± 0.46% after 50 μM MHME treatment (*p* < 0.001). Likewise, MHME at 50 μM triggered the increase in apoptotic populations of MDA-MB-231 and MDA-MB-436 cells from 7.36 ± 0.60% and 17.78 ± 0.86% to 52.90 ± 5.45% and 44.06 ± 0.50%, respectively (*p* < 0.001). These results clearly confirmed that MHME provokes apoptosis in TNBC cells, arguing that apoptotic death is a cause of MHME-induced cytotoxicity.

### 3.3. Blockade of STAT3 Activation Is Pivotal for MHME to Elicit TNBC Cytotoxicity

The mechanism whereby MHME induces TNBC cell apoptosis was explored. The STAT3 signaling is well-known for its pro-mitogenic and pro-survival activities, and, notably, deregulated STAT3 activation is a pivotal driver for TNBC initiation and progression [18]. Accordingly, we examined the effect of MHME on the STAT3 signaling activity. We found that, in all TNBC cell lines tested, MHME dose-dependently lowered the steady-state levels of tyrosine 705-phosphorylated STAT3 (p-STAT3), along with downregulation of BCL-2 protein, a canonical transcriptional target of STAT3 [32] (Figure 3A). Besides thwarting constitutive STAT3 activation, MHME blocked the STAT3 activation induced by external stimuli such as IL-6, as evidenced by the attenuation of an IL-6-elicited increase in p-STAT3 levels after MHME treatment (Figure 3B). These findings together identified MHME as an inhibitor of STAT3 activation.

With MHME recognized as a STAT3 blocker, we further evaluated the functional significance of STAT3 blockage for MHME to exert TNBC cytotoxicity. To this end, HS578T and MDA-MB-231 cells were engineered to withstand MHME’s inhibitory effect on STAT3 activation by stably expressing a dominant-active mutant of STAT3 (STAT3-C) [29]. The STAT3-C stable clones of HS578T and MDA-MB-231 cells were employed to address whether TNBC cells with sustained STAT3 activity are refractory to MHME-induced cytotoxicity. It is noteworthy that, although the vector control clones were susceptible to MHME’s proapoptotic action, MHME-induced apoptosis was abrogated in STAT3-C stable clones, as evidenced by the marked reduction in the levels of both PARP cleavage (Figure 3C) and Annexin V-positive populations (Figure 3D) in cells stably expressing STAT3-C. As shown in Figure 3D, the levels of apoptotic populations of HS578T cells were dropped from 81.17 ± 2.22% in vector control clones to 21.87 ± 1.42% in STAT3-C clones (*p* < 0.001); similarly, the extent of MDA-MB-231 apoptotic populations was 52.91 ± 5.45% in vector control clones, which was lowered to 32.52 ± 6.28% in STAT3-C stable clones (*p* < 0.01). Notably, the clonogenicity of STAT3-C stable clones was increased in parallel to the decrease in apoptosis after MHME treatment, supporting that apoptotic death is a primary cause of MHME-induced cytotoxicity (Figure 3E). Therefore, the finding that ectopically sustained STAT3 activation sabotages MHME’s proapoptotic/cytotoxic action underpins the notion of STAT3 blockade as a fundamental mechanism of action whereby MHME exerts its TNBC cytotoxicity.

### 3.4. MHME Blocks STAT3 to Downregulate BCL-2 for Inducing Apoptosis in TNBC Cells

We previously revealed that the decrease in p-STAT3 levels in MHME-treated TNBC cells is accompanied by a paralleled downregulation of BCL-2 (Figure 3A). Later, we found MHME failed to reduce BCL-2 protein levels in TNBC cells stably expressing STAT3-C, confirming that MHME downregulates BCL-2 by curbing STAT3 activation (Figure 4A). Given that BCL-2 is a potent antiapoptotic protein, we are curious whether BCL-2 downregulation accounts for the apoptosis elicited by MHME in TNBC cells. To address this question, we generated BCL-2 stable clones of HS578T and MDA-MB-231 cells to assess their sensitivity to MHME-induced apoptosis. We noticed MHME raised c-PARP levels in vector control clones while failing to promote PARP cleavage in cells stably expressing BCL-2 (Figure 4B). Along this line, MHME-induced upregulation of Annexin V-positive population was abrogated in TNBC cells where BCL-2 was not downregulated (*p* < 0.001) (Figure 4C). Thus, the data supports that BCL-2 downregulation is a crucial event downstream of STAT3 blockade to mediate MHME’s proapoptotic action on TNBC cells.

### 3.5. MHME Inhibits SRC to Repress STAT3 Activation in TNBC Cells

Last of all, we clarified how MHME represses STAT3 activation. The non-receptor tyrosine kinase SRC is well-known to directly phosphorylate the tyrosine 705 residue of STAT3 to induce STAT3 activation. Herein, we uncovered that MHME lowered the levels of active SRC (i.e., the tyrosine 416-phosphorylated SRC (p-SRC)) in all tested TNBC cell lines (Figure 5A), suggesting MHME impairs SRC activation. To validate the causative role of SRC inhibition in MHME-induced STAT3 blockage, we addressed whether MHME’s inhibitory effect on STAT3 activation in TNBC cells is sabotaged when SRC activity is sustained through ectopic expression of *v-src*, which encodes a dominant-active src [30]. Notably, immunoblotting revealed that MHME failed to downregulate p-STAT3 in the *v-src* stable clones of HS578T and MDA-MB-231 cells, in contrast to the decreased p-STAT3 levels in their vector control clones after MHME treatment (Figure 5B). Moreover, consistent with its effect to sustain STAT3 activation, *v-src* ectopic expression rescued TNBC cells from MHME-induced apoptotic death, as evidenced by the resistance of *v-src* stable clones to MHME-triggered increase in the levels of c-PARP (Figure 5B) and Annexin V-positive populations (Figure 5C). Altogether, these results delineate that SRC inhibition is responsible for the blockade of STAT3 activation in MHME-treated TNBC cells.

## 4. Discussion

In this study, we used a panel of human TNBC cell lines, including HS578T, MDA-MB-231, and MDA-MB-436 cells, as the cell model to demonstrate the in vitro anti-TNBC effect of MHME and the underlying cytotoxic mechanism of action for the first time. We started by showing the cytotoxic activity of MHME on TNBC cells, which is more potent than Hispolon’s (Figure 1). We then found that MHME engages apoptosis to eliminate TNBC cells (Figure 2). In addition, we revealed that MHME thwarts the STAT3-mediated signaling, as evidenced by MHME’s inhibitory action on both constitutive and IL-6-inducible activation of STAT3, and, notably, the blockade of STAT3 activation was validated as a pivotal mechanism of MHME’s proapoptotic/cytotoxic action on TNBC cells (Figure 3). We further proved that, as a result of STAT3 blockage, MHME downregulates BCL-2 to provoke apoptosis in TNBC cells, hence reinforcing the blockade of STAT3 signaling as a vital proapoptotic/cytotoxic mechanism of MHME on TNBC cells (Figure 4). Lastly, we elucidated that MHME impairs the activation of STAT3 by suppressing SRC activation (Figure 5). To our best knowledge, the current findings of MHME’s in vitro anti-TNBC effect and the fundamental role of the blockade of the SRC/STAT3/BCL-2 axis in MHME’s proapoptotic/cytotoxic action on TNBC cells have never been documented previously.

In contrast to the intensive studies on Hispolon’s anticancer effect, research about the anticancer effect of Hispolon analogs with the underlying mechanisms remains limited. Along this line, the current report of MHME’s in vitro anti-TNBC effect, together with our previous discoveries on the in vitro cytotoxic action of HME and DHME on bladder and colorectal cancers, respectively [25,26,27], highlights the potential to translate MHME and other Hispolon analogs into novel anticancer agents. Therefore, it would be interesting to profile the types of human cancers susceptible to the cytotoxic effect of Hispolon analogs in the future.

Current results argue that induction of apoptotic death appears as a primary cause of MHME’s cytotoxic action on TNBC cells. Still, we do not exclude the likely involvement of additional cell-death modes, such as autophagy and ferroptosis, in MHME-induced TNBC cytotoxicity. It should be noted that previous studies have revealed that Hispolon promotes the induction of autophagy in MDA-MB-231 cells [33] and cervical cancer cell lines HeLa and SiHa [34]. Although whether MHME is pro-autophagic remains elusive, it would be interesting to clarify whether and how autophagy contributes to MHME’s cytotoxic action on TNBC cells in the future.

Data presented here underscores the blockade of the STAT3-mediated signaling axis as a pivotal mechanism of MHME’s proapoptotic action on TNBC cells. Besides MHME, it is noteworthy that we previously demonstrated that HME and DHME target the STAT3-mediated pro-survival signaling to slay bladder and colorectal cancer cells, respectively [25,27]. Furthermore, in a prostate cancer cell line DU145, Hispolon-induced apoptosis was accompanied by a paralleled decrease in p-STAT3 levels [35]. Based on the above evidence, it is plausible to speculate that the STAT3 signaling pathway is a common target of suppression for Hispolon-based compounds. Future investigations should confirm the inhibitory effect of MHME and other Hispolon analogs on the STAT3 activity in a broad range of human cancers to address this speculation.

It is generally recognized that the limited treatment options for TNBC are due to the shortage of molecular markers for targeted therapies [18]. An alternative approach to circumvent this issue is to target the aberrantly activated signaling pathways fundamental to sustaining TNBC development and progression [11,12,13,14]. Considering the essential role of deregulated STAT3 activation in driving TNBC genesis, progression, and chemoresistance, STAT3 has emerged as a foremost TNBC drug target [18]. In line with this, our discovery of MHME functioning as an inhibitor of STAT3 activation holds promising implications in TNBC treatment. 

One of the burning questions to be resolved is how MHME inhibits STAT3 activation in TNBC cells, as revealed by the decrease in tyrosine 705-phosphorylated STAT3 levels after MHME treatment (Figure 3). In principle, there are several points of interference along the STAT3 signaling pathway where MHME might engage to downregulate STAT3 tyrosine 705 phosphorylation [18]. For instance, MHME could target the upstream regulators of STAT3 by suppressing the activity of STAT3 upstream kinases like JAKs or SRC. Alternatively, MHME might upregulate tyrosine phosphatases like SHP-1 or SHP-2 to downregulate STAT3 phosphorylation. Moreover, MHME could directly bind to STAT3 to impede its phosphorylation, as previously demonstrated by the anti-TNBC effects of several natural compounds like Alantolactone [36] and Arctigenin [37]. In this report, we verified the inhibition of SRC, one of the upstream kinases of STAT3, as a mechanism employed by MHME to downregulate STAT3 tyrosine 705 phosphorylation (Figure 5). Still, it is more informative to elucidate whether MHME suppresses STAT3 activation by modulating the expression of SHP-1/SHP-2 or through direct binding to STAT3 in our follow-up studies.

Another intriguing discovery of this study is that MHME acts as an inhibitor of SRC activation (Figure 5). Growing evidence has unraveled that oncogenic overexpression or dysregulated activation of SRC drives breast cancer development and progression [38]. Notably, the potential of SRC as a TNBC drug target was underpinned by several preclinical studies demonstrating that pharmacological blockade of SRC activity confers therapeutic benefits on TNBC, particularly the TNBC cells with vimentin overexpression [39] or displaying a characteristic phosphotyrosine signature [40]. Accordingly, the evidence that MHME inhibits SRC besides STAT3 further reinforces MHME’s therapeutic potential on TNBC.

The present in vitro evidence uncovers that MHME is cytotoxic to TNBC cells by thwarting the SRC/STAT3/BCL-2 pro-survival axis. Still, it should be noted that several limitations of the current study remain to be resolved in our follow-up studies. First, the concern about whether MHME selectively kills malignant while sparing normal breast epithelial cells must be addressed. Second, to reinforce the potential of applying MHME to TNBC’s chemotherapy regimens, murine models of TNBC must be employed to validate tumor growth retardation and p-STAT3 downregulation in tumor-planted mice under MHME administration. Finally, toxicological and pharmacokinetic analyses in MHME-treated animals must be performed to acquire information about the maximum and average dosages of MHME achieved in animals and humans.

In conclusion, we herein unraveled the in vitro anti-TNBC effect of MHME for the first time. Mechanistically, MHME slays TNBC cells by thwarting the SRC/STAT3/BCL-2 pro-survival axis to provoke TNBC cell apoptosis (Figure 6). Our discovery suggests the potential of MHME to be regarded as a TNBC therapeutic agent, either administrated alone or combined with current chemo- or immunotherapeutics.

## Figures and Tables

**Figure 1 biomedicines-11-02742-f001:**
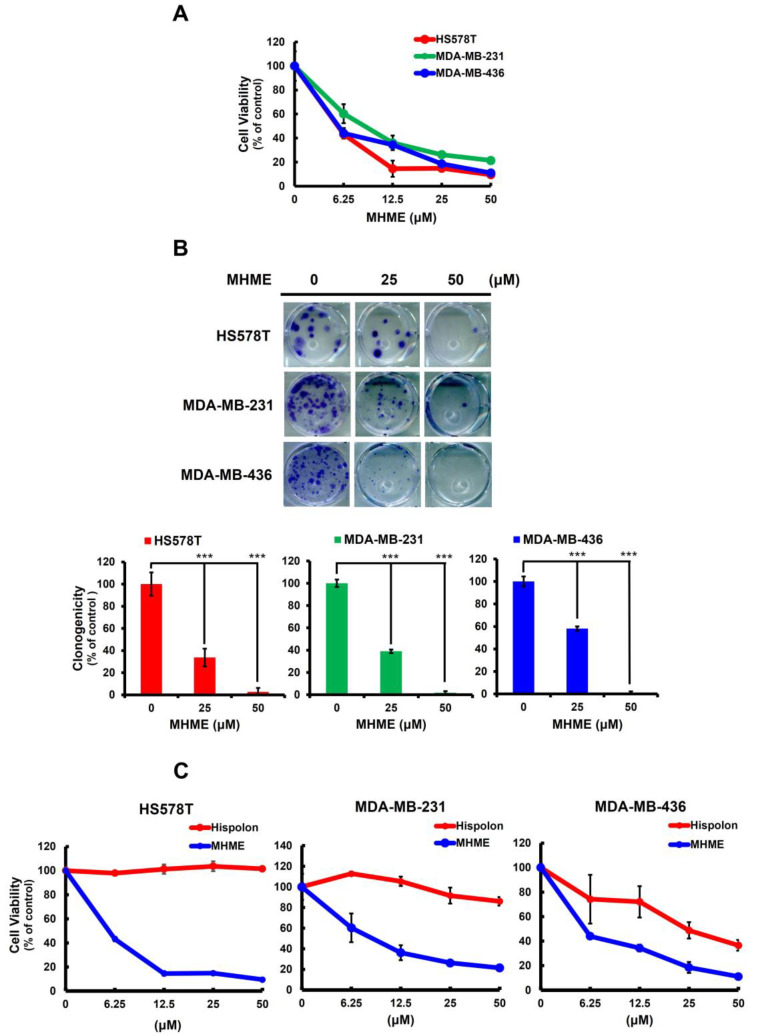
MHME’s TNBC cytotoxic effect. (**A**) *MHME elicits* in vitro *cytotoxicity against TNBC cells*. Human TNBC cell lines, including HS578T, MDA-MB-231, and MDA-MB-436 cells, were subject to 48 h treatment with various concentrations of MHME (0, 6.25, 25, 50 μM) for subsequent determination of cell viability using MTS assay. (**B**) *MHME impedes TNBC cells to form colonies*. TNBC cell lines were treated with MHME (0, 25, 50 μM), followed by growing in drug-free media for 14 d to form colonies. (**C**) *The TNBC cytotoxic effect of MHME is more potent than that of Hispolon*. HS578T, MDA-MB-231, and MDA-MB-436 cells were treated for 48 h with a graded dosage of MHME or Hispolon, followed by an MTS assay for cell viability. *** *p* < 0.001.

**Figure 2 biomedicines-11-02742-f002:**
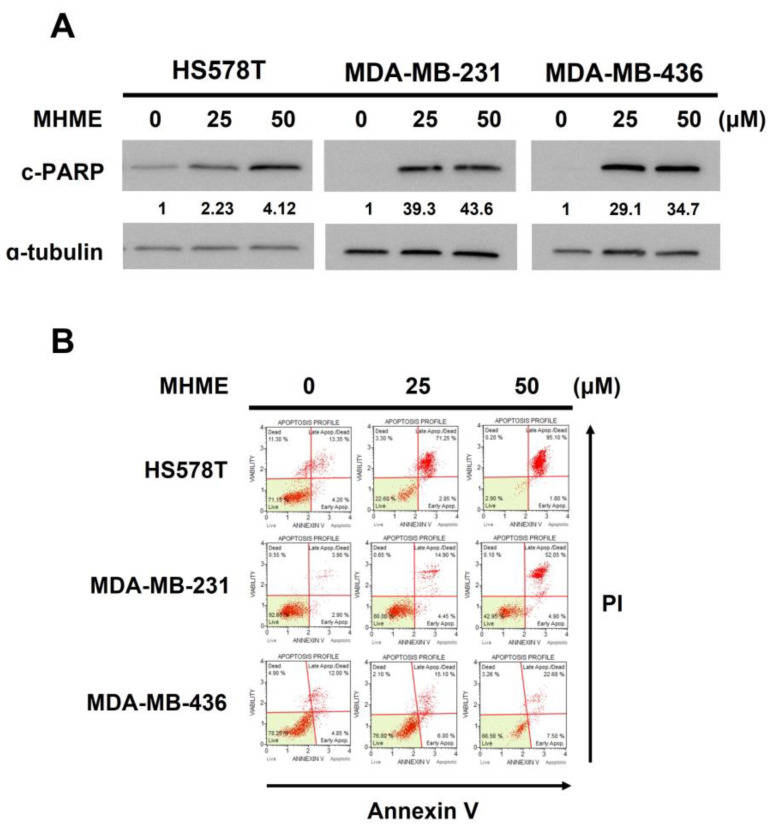
MHME induces apoptosis in TNBC cells. HS578T, MDA-MB-231, and MDA-MB-436 cells after 24 h treatment with MHME (0, 25, 50 μM) were subject to evaluation for the induction of apoptosis, which was revealed by (**A**) the upregulation of cleaved PARP (c-PARP), an apoptosis marker, using immunoblotting and by (**B**,**C**) the increased levels of Annexin V-positive (apoptotic) cell populations using flow cytometry analysis. *** *p* < 0.001.

**Figure 3 biomedicines-11-02742-f003:**
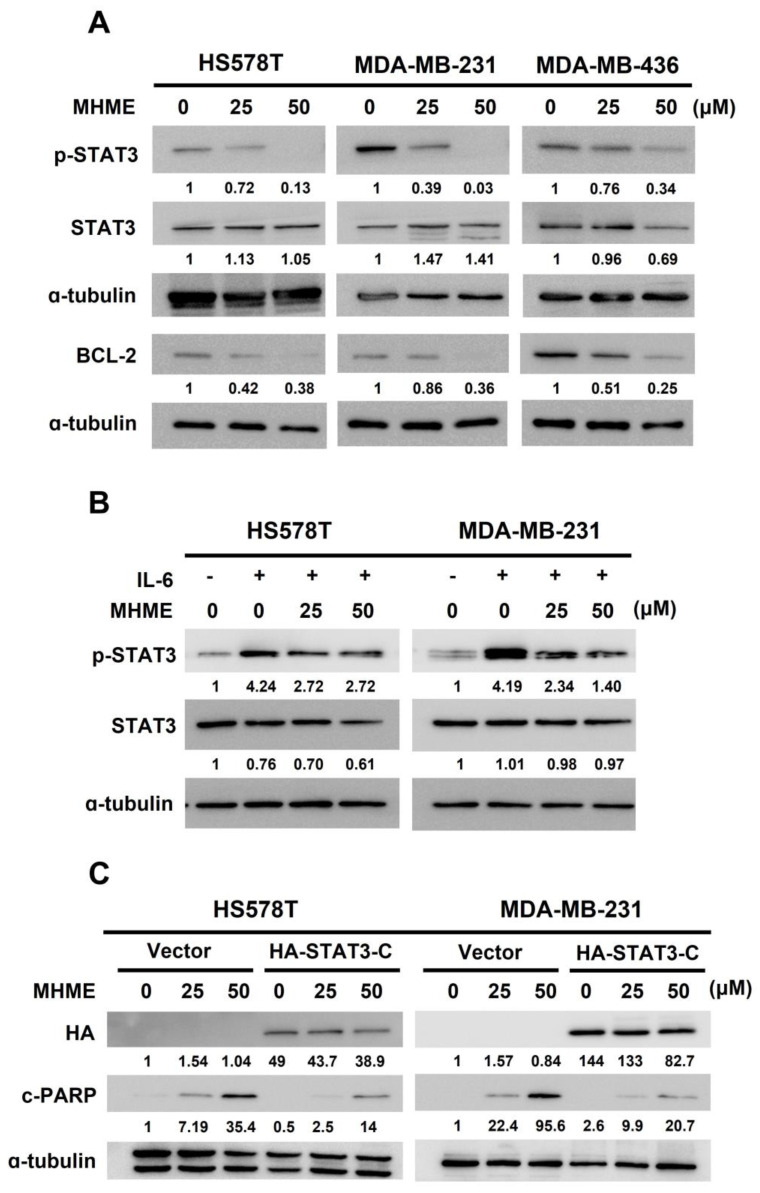
STAT3 blockage is essential for MHME to induce TNBC cytotoxicity. (**A**) *MHME suppresses constitutive STAT3 activation in TNBC cells*. HS578T, MDA-MB-231, and MDA-MB-436 cells were treated with MHME (0, 25, 50 μM) for 24 h, followed by immunoblotting for the levels of tyrosine 705-phosphorylated STAT3 (p−STAT3), total STAT3, and BCL-2. α-tubulin levels were used as the equal loading control. (**B**) *MHME represses IL-6*−*inducible STAT3 activation in TNBC cells*. HS578T and MDA-MB-231 were pre−treated with IL-6 (100 ng/mL) for 30 min, followed by MHME treatment (25 or 50 μM) for 24 h and immunoblotting thereafter for the levels of p−STAT3 and total STAT3. α-tubulin levels were used as the equal loading control. (**C**,**D**) *Mitigation of MHME’s proapoptotic effect on the TNBC cells by sustained STAT3 activation*. HS578T and MDA-MB-231 cells stably expressing a dominant−active STAT3 mutant (STAT3 (A661C/N663C); STAT3−C) and the corresponding vector controls were treated with MHME (0, 25, 50 μM) for 24 h, followed by (**C**) immunoblotting for the levels of the hemagglutinin epitope (HA) and cleaved PARP (c−PARP) or by (**D**) flow cytometry analysis for the levels of Annexin V−positive populations to evaluate the extent of MHME−induced apoptosis. (**E**) *Restoration of clonogenicity in MHME-treated TNBC cells when STAT3 activation is sustained*. Clonogenicity assays were conducted in MHME-treated TNBC vector or STAT3−C stable clones to assess MHME-induced cytotoxicity. ** *p* < 0.01. *** *p* < 0.001.

**Figure 4 biomedicines-11-02742-f004:**
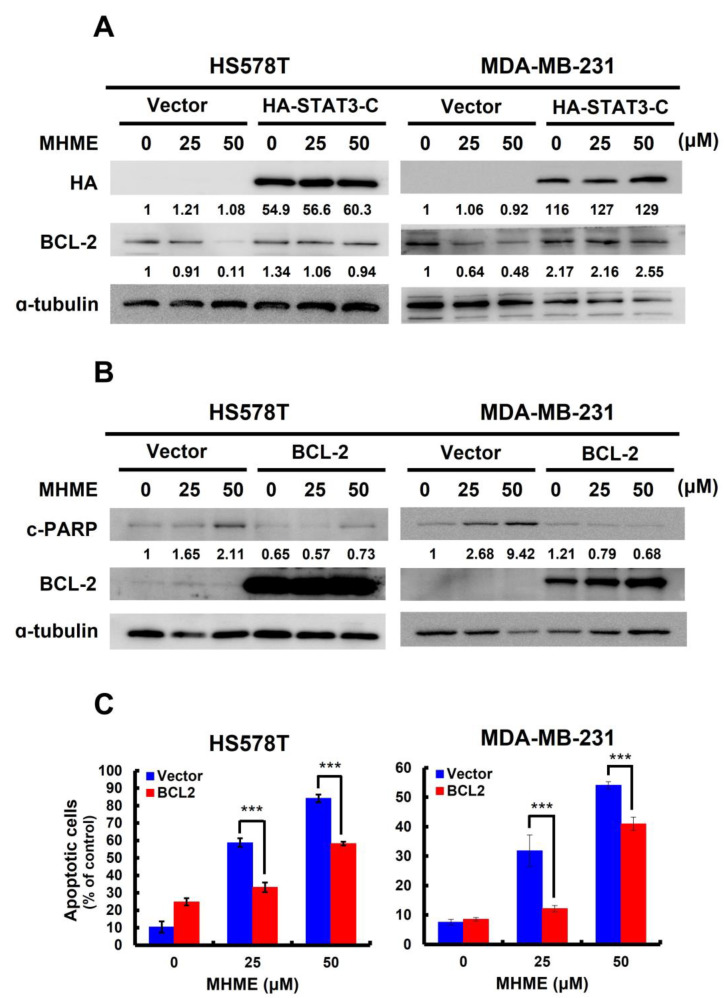
STAT3 blockage underlies MHME-induced downregulation of BCL-2 to evoke TNBC cell apoptosis. (**A**) *MHME-induced downregulation of BCL-2 depends on STAT3 blockage*. The STAT3-C stable clones of HS578T and MDA-MB-231 cells and their vector controls were subject to MHME treatment (0, 25, 50 μM) for 24 h, followed by immunoblotting for the levels of HA and BCL-2. (**B**,**C**) *BCL-2 overexpression curtails MHME’s proapoptotic action on TNBC cells*. BCL-2 stable clones of HS578T and MDA-MB-231 cells and their vector controls after 24 h treatment with MHME (0, 25, 50 μM) were assessed for the induction of apoptosis by c-PARP immunoblotting (**B**) and the amounts of Annexin V-positive (apoptotic) cell population using flow cytometry analysis (**C**). α-tubulin levels were used as the equal loading control in immunoblotting. *** *p* < 0.001.

**Figure 5 biomedicines-11-02742-f005:**
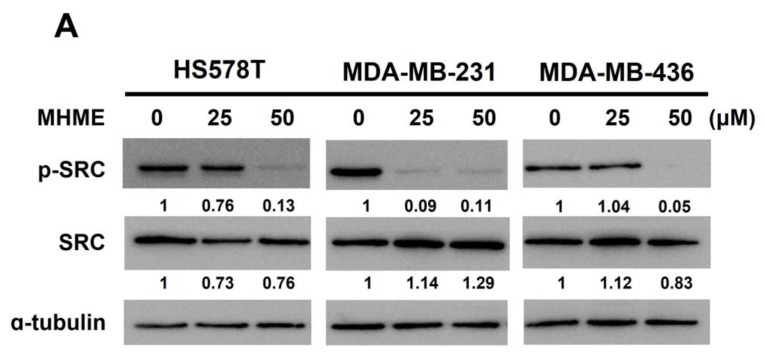
SRC inhibition accounts for MHME-induced STAT3 blockage in TNBC cells. (**A**) *MHME impairs SRC activation*. Human TNBC cells after 24 h treatment with MHME (0, 25, 50 μM) were subject to immunoblotting for the amounts of tyrosine 416-phosphorylated SRC (p-SRC), a surrogate marker of SRC activation, along with unphosphorylated SRC. (**B**) *Persistent SRC activation sabotages MHME-induced STAT3 blockage*. The stable clones of *v-src* (a dominant-active SRC mutant) or its vector control clones of HS578T and MDA-MB-231 cells were treated with MHME for 24 h, followed by immunoblotting for the levels of p-STAT3 (Y705) and cleaved PARP (c-PARP). (**C**) *Persistent SRC activation lessens MHME-induced apoptosis*. The v-src stable clones and their vector control clones of HS578T and MDA-MB-231 cells were examined for the amounts of Annexin V-positive (apoptotic) cell populations after 24 h treatment with MHME. α-tubulin levels in immunoblotting were used as the equal loading control. *** *p* < 0.001.

**Figure 6 biomedicines-11-02742-f006:**
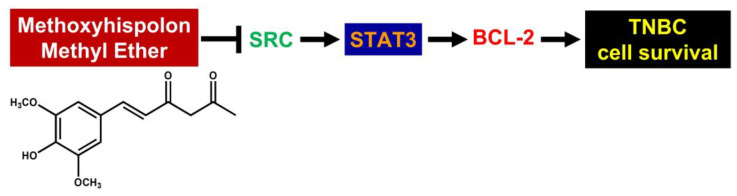
Schematic diagram of MHME’s mechanism of cytotoxic action on TNBC cells revealed in the current study. MHME slays TNBC cells by blocking SRC-mediated STAT3 activation to downregulate BCL-2, leading to the induction of TNBC cell apoptosis. The chemical structure of MHME is adapted from Ravindran et al. [21].

## Data Availability

Data will be made available by the corresponding author (chiachechang@gmail.com) upon reasonable request.

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
