# Peer review of "Methoxyhispolon Methyl Ether, a Hispolon Analog, Thwarts the SRC/STAT3/BCL-2 Axis to Provoke Human Triple-Negative Breast Cancer Cell Apoptosis In Vitro"

_biomedicines, 2023, doi:10.3390/biomedicines11102742_

Round 1
Reviewer 1 Report
A promising TNBC treatment approach is targeting the oncogenic signaling pathways pivotal to TNBC initiation and progression. Deregulated activation of signal transducer and activator of transcription 3 (STAT3) is fundamental to driving TNBC malignant transformation, highlighting STAT3 as a promising TNBC therapeutic target. Methoxyhispolon Methyl Ether (MHME) is an analog of Hispolon, an anti-cancer polyphenol found in the medicinal mush-room Phellinus linteus.
Collectively, the authors conclude that MHME provokes TNBC cell apoptosis through the blockade of the SRC/STAT3/BCL-2 prosur-vival axis. Their findings implicate the potential of applying MHME as a TNBC chemotherapy agent.
In conclusion, the authors unraveled the in vitro anti-TNBC effect of MHME for the first time. Mechanistically, MHME slays TNBC cells by thwarting the SRC/STAT3/BCL-2 prosurvival axis to provoke TNBC cell apoptosis. Our discovery implicates the potential to translate MHME into a novel anticancer agent for TNBC therapy, either administrated alone or combined with current chemo- or immunotherapeutics.
The approach is innovative and therefore, based on its notable scientific contribution, I consider that it can be accepted for publication. The figures have adequate statistical rigor and the bibliographic references are updated.
As an improvement, I would say that the chemical structure of methoxyhispolon Methyl Ether must be provided.
The English is O.K. and only minor mistakes must be corrected.
Reviewer 2 Report
Dear authors
You test a polyphenol and a derivative against TNBC cells. However, I could not find in your manuscript any control regarding normal cells. Therefore, we do not know if this polyphenolic derivative is a general poison or is a specific cytotoxic for TNBC without affecting normal cells. None of the figures include normal controls in case you did them.
So, you show that MHME kills TNBC cells, but we do not know if it kills normal cells as well.
Another missing data in your manuscript is about the maximum and average concentration that this polyphenol derivative can achieve in humans or animals. While you do not have this information there is no way to know if it can have any importance in cancer treatment.
Hispolon has been identified as a STAT3 inhibitor before your manuscript was delivered. Please see your own reference 35:
Masood M, Rasul A, Sarfraz I, Jabeen F, Liu S, Liu X, Wei W, Li J, Li X. Hispolon induces apoptosis against prostate DU145 cancer cells via modulation of mitochondrial and STAT3 pathways. Pak J Pharm Sci. 2019 Sep;32(5(Supplementary)):2237-2243. PMID: 31894049.
It would also be important to compare apoptotic effects of the hispolon derivative with classic chemotherapeutic drugs such as cisplatin.
English is acceptable.
Round 2
Reviewer 2 Report
no comments
I repeat the suggestions expressed in the first round and which have not been modified in this corrected version.
If they do not want to correct their English, that is fine. After all, Bad English is the most frequently used language nowadays.